# Lectin-Like Transcript 1 (LLT1) Checkpoint: A Novel Independent Prognostic Factor in HPV-Negative Oropharyngeal Squamous Cell Carcinoma

**DOI:** 10.3390/biomedicines8120535

**Published:** 2020-11-25

**Authors:** Mario Sanchez-Canteli, Francisco Hermida-Prado, Christian Sordo-Bahamonde, Irene Montoro-Jiménez, Esperanza Pozo-Agundo, Eva Allonca, Aitana Vallina-Álvarez, César Álvarez-Marcos, Segundo Gonzalez, Juana M. García-Pedrero, Juan P. Rodrigo

**Affiliations:** 1Department of Otolaryngology, Hospital Universitario Central de Asturias, Instituto de Investigación Sanitaria del Principado de Asturias, 33011 Oviedo, Spain; mariosanchezcanteli@gmail.com (M.S.-C.); franjhermida@gmail.com (F.H.-P.); imj21897@gmail.com (I.M.-J.); espe6196@hotmail.com (E.P.-A.); ynkc1@hotmail.com (E.A.); caalvarez@uniovi.es (C.Á.-M.); 2Instituto Universitario de Oncología del Principado de Asturias, University of Oviedo, 33006 Oviedo, Spain; Christiansbl87@gmail.com (C.S.-B.); alaicla@hotmail.es (A.V.-Á.); segundog@uniovi.es (S.G.); 3CIBERONC, Instituto de Salud Carlos III, 28029 Madrid, Spain; 4Department of Functional Biology, Instituto de Investigación Sanitaria del Principado de Asturias, University of Oviedo, 33006 Oviedo, Spain; 5Department of Pathology, Hospital Universitario Central de Asturias, ISPA, 33011 Oviedo, Spain

**Keywords:** lectin-like transcript 1, LLT1, CLEC2D, immune checkpoint, NK cell, head and neck squamous cell carcinomas, human papillomavirus, prognosis

## Abstract

Lectin-like transcript 1 (LLT1) expression by tumor cells contributes to immune evasion, thereby emerging as a natural killer (NK) cell-mediated immunotherapeutic target. This study is the first to investigate LLT1 expression (encoded by *CLEC2D* gene) in head and neck cancers to ascertain its impact on patient prognosis. LLT1 expression was analyzed by immunohistochemistry in a homogeneous cohort of human papillomavirus (HPV)-negative oropharyngeal squamous cell carcinomas (OPSCC), and correlated with clinical data. Results were further validated using transcriptomic data from the TCGA database. Tumoral LLT1 expression was detected in 190/221 (86%) OPSCC specimens, whereas normal pharyngeal epithelium was negative. Patients harboring LLT1-positive tumors showed significantly lower disease-specific (DSS) and overall survival (OS) (*p* = 0.049 and *p* = 0.036, respectively, log-rank test). High density of LLT1-positive tumor-infiltrating lymphocytes (TIL) was also frequently detected in 160 (73%) OPSCC samples, and significantly associated with better DSS and OS (*p* < 0.001 and *p* = 0.007, respectively). Multivariate Cox analysis further revealed that tumoral LLT1 expression and infiltration of LLT1-positive TIL were independent prognostic factors for DSS and OS. *CLEC2D* mRNA levels are also significantly increased in primary tumors compared to normal tissue. Strikingly, the prognostic impact of *CLEC2D* mRNA levels varied depending on HPV status in OPSCC, and among distinct cancer types. CLEC2D expression was significantly correlated with NK cell infiltration using the MCP-counter model. These findings uncover LLT1/*CLEC2D* as an independent prognostic factor in HPV-negative OPSCC, and a potential novel target for immunotherapy.

## 1. Introduction

Head and neck squamous cell carcinoma (HNSCC) is the most frequent malignancy in the head and neck region, and the sixth most commonly diagnosed cancer [1]. Despite the latest advances in treatment, oncological outcomes remained poor and barely improved over the last decades. Therefore, new biomarkers and/or therapeutic targets are needed to improve the prognosis of these neoplasms.

HNSCC is a highly complex disease that arises at various anatomical locations (oral cavity, oropharynx, hypopharynx, and larynx), thereby widely differing in tumor biology, etiology (tobacco and/or HPV infection), clinical behavior, and response to treatment [2]. Taking this into consideration, the study of these tumors should involve homogeneous cohorts of patients; therefore, it is advisable to focus on specific tumor locations and similar etiologic factors [3].

Evasion of the immune system is an important hallmark of cancer and various mechanisms of immune evasion have been characterized and become key targets for anti-cancer immunotherapy [4]. Among immune cells, natural killer (NK) cells, which function within the innate immune system, play a major role in anti-tumor and antiviral responses. Killer cell lectin-like receptors (KLRs) are C-type lectin-like glycoproteins encoded by genes clustered in the natural killer gene complex (NKC) located on the short arm of human chromosome 12 [5,6,7]. In addition to the NKG2 subfamily, the NKC includes a less characterized group of genes coding for NKRP1 (CD161) receptors and their ligands of the C-type lectin (CLEC) subfamily. Among this group, the best recognized is the NKRP1A/LLT1 pair encoded by the *KLRB1* and *CLEC2D* genes, respectively [8]. Both molecules are type II transmembrane-signaling glycoproteins with an extracellular C-type lectin domain. NKRP1A is predominantly expressed on NK cells, where it acts as an inhibitory receptor.

Expression of LLT1 by tumor cells may facilitate their escape from NK cell surveillance [9]. On the other hand, NKRP1A and LLT1 may modulate the activity of NK cells, T and B lymphocytes in inflammatory processes and also the pathogenesis of autoimmune disorders [10,11]. Thus, the NKRP1A/LLT1 receptor/ligand system could emerge as a potential new therapeutic target for cancer treatment as well as some autoinflammatory disorders [5].

A detailed analysis of LLT1 expression pattern and distribution in human tissue is lacking. LLT1 expression was found in circulating B cells and monocytes, but not in lung and liver-resident macrophages. Strikingly, high LLT1 expression was detected in immune-privileged sites, such as the brain, placenta and testes, where it was confirmed the ability of LLT1 to inhibit NK cell function [12].

Accordingly, since LLT1 is expressed in different cell and tissue types and it could facilitate cancer cells to escape from immunosurveillance, a possible role in cancer progression has been postulated. In this sense, it has been shown that LLT1 expression in triple-negative breast cancer cells and prostate cancer cells inhibits NK cell response [13]. However, the expression and clinical significance of LLT1 in HNSCC has not yet been investigated.

The overall goal of our study was to analyze LLT1 expression pattern in a large homogeneous cohort of HNSCC patients from a single subsite (oropharynx) and with known HPV status, and to establish associations with clinic and pathologic features and the possible impact on patient’s prognosis.

## 2. Results

### 2.1. Immunohistochemical Analysis of LLT1 Expression in Oropharyngeal Squamous Cell Carcinomas (OPSCC) Tissue Specimens

LLT1 immunostaining was successfully evaluated in 221 (92%) out of 241 HPV-negative OPSCC; tumor samples with inadequate tissue integrity and/or tumor representability were excluded from the evaluation. The clinicopathological characteristics of the OPSCC cohort are shown in Table 1.

190 (86%) of the 221 OPSCC samples exhibited positive LLT1 expression in the tumor cells: 144 (65%) showed weak to moderate expression and 46 (21%) strong expression (Figure 1). Normal pharyngeal epithelium showed negative LLT1 expression, whereas strong expression was observed in tumor-infiltrating lymphocytes with variable abundance.

### 2.2. Correlations of LLT1 Expression with Clinicopathological Parameters

Positive LLT1 expression was more frequent in patients with tumor recurrence and those who died from the tumor, although the differences did not reach statistical significance. No correlations with other clinical and pathological characteristics were observed (Table 1).

### 2.3. Evaluation of LLT1 Protein Expression in the Intratumor Immune Microenvironment

The role of LLT1 expression was also assessed in the intratumor immune infiltrate. To accomplish this, LLT1 immunostaining in the tumor-infiltrating lymphocytes (TIL) was successfully evaluated in 220 HPV-negative OPSCC tumor specimens and correlated with tumoral LLT1 expression. Representative images for the different LLT1 expression scores are shown in Figure 2. 8 tumors showed absence or sporadic LLT1-positive TIL (scored as 0), 52 mild to moderate LLT1-positive TIL (scored as 1), 89 abundant, and 71 highly abundant LLT1-positive TIL (scored as 2 and 3, respectively). Therefore, 60 (27%) cases showed low density of LLT1-positive TIL and 160 (73%) high LLT1-positive TIL infiltration. In addition, LLT1 expression in TILs was not correlated with LLT1 expression in tumor cells (Spearman’s Rho correlation coefficient = 0.033, *p* = 0.625).

### 2.4. Impact of LLT1 Expression on OPSCC Survival

Patients harboring LLT1-positive tumors showed a significantly lower disease-specific survival (DSS) (*p* = 0.049, log-rank test; Figure 3A) and overall survival (OS) (*p* = 0.036, log-rank test; Figure 3B). However, no differences in survival rates were observed between the different levels of LLT1 expression (i.e., low, moderate vs. strong staining, *p* = 0.94 for DSS and *p* = 0.43 for OS, log-rank test) (data not shown).

By contrast, patients harboring tumors with high infiltration of LLT1-positive TIL exhibited significantly better DSS (*p* < 0.001, log-rank test; Figure 3C) and OS (*p* = 0.007, log-rank test; Figure 3D). Moreover, DSS and OS was found to gradually increase with the density of LLT1-positive TIL infiltration (*p* < 0.001, log-rank test).

We next assessed the impact on prognosis of the combination of both features LLT1 expression in the tumor cells and infiltration of LLT1-positive TIL. We found that patients with LLT1-negative tumors and high density of LLT1-positive TIL (LLT1tum −/LLT1 TIL high) showed the highest survival rates, whereas those patients with LLT1-positive tumors and low density of LLT1-positive TIL (LLT1tum +/LLT1 TIL low) had the worst prognosis, with the other combinations showing intermediate survival rates (*p* = 0.001 for DSS and *p* = 0.017 for OS, log-rank test; Figure 3E,F).

Furthermore, multivariate Cox analysis, including pT classification (T1-T2 vs. T3-T4), pN classification (N0 vs. N+), degree of differentiation, tumoral LLT1 expression (positive vs. negative), and infiltration of LLT1-positive TIL (low vs. high) showed that the parameters independently associated with a worse DSS were T3-T4 classification (HR = 1.56, 95% CI = 1.04–2.33, *p* = 0.029), N+ classification (HR = 2.08, 95% CI = 1.28–3.37, *p* =0.003), positive tumoral LLT1 expression (HR = 1.79, 95% CI = 1.01–3.19, *p* = 0.04), and low infiltration of LLT1-positive TILs (HR = 1.75, 95% CI = 1.21–2.53, *p* = 0.003). Similarly, the parameters independently associated with a worse OS were: T3-T4 classification (HR = 1.51, 95% CI = 1.07–2.11, *p* = 0.017), N+ classification (HR = 1.68, 95% CI = 1.15–2.46, *p* = 0.007), and low infiltration of LLT1-positive TILs (HR = 1.45, 95% CI = 1.03–2.03, *p* = 0.031).

### 2.5. In Silico Analysis of mRNA Levels Using The Cancer Genome Atlas (TCGA) HNSCC Database

The expression and prognostic relevance of *CLEC2D* gene, encoding LLT1 protein, was also investigated using transcriptomic data from the TCGA cohort of 523 HNSCC patients [14]. In good agreement with our data on LLT1 protein, *CLEC2D* mRNA levels were found to significantly increase in primary tumors compared to normal tissue samples (*p* < 0.001, Figure 4A). Moreover, patients with HPV-negative OPSCC harboring high *CLEC2D* mRNA levels exhibited a significantly lower OS (*p* < 0.001, log-rank test) (Figure 4B). By contrast, high *CLEC2D* mRNA levels were significantly associated with a better prognosis in HPV-positive OPSCC patients (*p* < 0.001, log-rank test) (Figure 4C).

The analysis of *CLEC2D* expression and patient survival among different cancers using the comprehensive web server resource TIMER 2.0 [15] further revealed striking differences on prognosis (Figure 5A). *CLEC2D* expression was strongly and significantly associated with a poor prognosis in specific HNSCC subtypes (i.e., HPV-negative HNSCC) and kidney carcinomas (i.e., KIRC and KICH). In marked contrast, *CLEC2D* expression showed significant association with a good prognosis in HPV-positive HNSCC and skin cutaneous melanoma (SKCM) among others (Figure 5A). The MCP-counter model was used to estimate the abundance of tumor-infiltrating NK cells in relation to *CLEC2D* expression [16]. A strong correlation between *CLEC2D* levels and NK cell infiltration was observed in HPV-negative HNSCC (Figure 5B). When assessing the impact on patient survival of these two features combined, we found that the subset of patients harboring high *CLEC2D* expression (*CLEC2D*^high^) and low infiltration of NK cells exhibited the lowest survival rates, in good agreement with our results at protein level (Figure 5C). The gene set enrichment analysis (GSEA) revealed different subsets of genes significantly overexpressed in *CLEC2D*^high^ tumors, which are involved in immune response and various tumor-associated biological processes (Figure 6). By contrast, gene sets significantly overexpressed in *CLEC2D*^low^ tumors were related to epithelial development and regulation of cell polarity (Figure 6).

## 3. Discussion

Different mechanisms of tumor immune evasion have been described in HNSCC, and various immune checkpoints are found overexpressed in the surrounding tumor microenvironment to induce the exhaustion of effector T cells, being the PD-1/PD-L1 the most extensively studied [4,17]. In marked contrast, only few studies have so far investigated the role of LLT1/NKRP1(CD161) inhibitory checkpoint in cancer. In a recent metadata study of 18,000 tumors across 39 malignancies, Gentles et al. [18] reported KLR1B (encoding CD161) expression as the most favorable prognostic marker, reflecting a favorable prognostic T cell signature as a marker of enhanced innate immune characteristics. In agreement with these results, in a meta-analysis of 730 publicly-available transcriptomic profiles of non-small cell lung cancer (NSCLC) specimens, Braud et al. [19] found a positive association of CLEC2D (encoding LLT1) and KLRB1 (encoding CD161) gene expression with favorable outcome. Interestingly, in that study, LLT1 expression was restricted to immune cells within NSCLC tumor tissue. However, Matthew et al. [13] reported LLT1 overexpression in primary prostate tumors and carcinoma-derived cell lines. Moreover, disruption of LLT1 interaction with NKRP1A using an anti-LLT1 blocking antibody was found to increase the NK-mediated cytotoxicity of prostate cancer cells [13]. These data suggest a key role for LLT1-NKRP1A complex in immune evasion, and as such LLT1 expression in prostate cancer cells could contribute to the inhibition of NK cell-mediated cytolytic activity. Similarly, Marrufo et al. [20] detected LLT1 expression in triple negative breast cancer (TBNC) cell lines and demonstrated that blocking LLT1-NKRP1A complex on these cells using antibodies or LLT1 gene knockdown enhanced NK cell-mediated lysis of TNBC cells. In light of these findings, targeting LLT1-NKRP1A interaction has been proposed as a novel immunotherapeutic strategy to effectively treat TNBC and prostate cancer [21].

In agreement with these studies, we detected LLT1 expression in a high percentage of OPSCC samples, and positive LLT1 expression within the tumor cells was independently associated with poorer survival outcomes, suggesting a possible immunosuppressive effect of LLT1 expression. These results are in line with our previous observations in cutaneous squamous cell carcinomas, revealing LLT1 expression as a significant predictor of risk for nodal metastasis and reduced disease-specific survival [22]. These data suggest that the prognostic value of LLT1 expression may vary according to the histological type, being unfavorable in squamous carcinomas. In addition, this study also provides evidence for the role of LLT1 expression in the tumor immune infiltrate in our cohort of OPSCC patients. Thus, high density of LLT1-positive TIL was frequently detected, and significantly and independently associated with a better prognosis. These results are similar to those reported in NSCLC [19], indicating that LLT1 expression in the intratumor immune microenvironment may confer favorable outcomes in different tumor types.

Subsequent analysis of *CLEC2D* gene expression using TCGA database further confirmed our results on LLT1 protein. Accordingly, *CLEC2D* expression significantly increased in primary tumors compared to normal tissue, and high *CLEC2D* mRNA levels were strongly associated with lower survival rates in HPV-negative HNSCC patients. However, strikingly, *CLEC2D* mRNA levels were significantly associated with a more favorable prognosis specifically in the HPV-positive HNSCC subgroup. Moreover, the impact of *CLEC2D* expression on patient prognosis varied among different tumor subtypes, with either favorable or unfavorable outcomes depending on the cancer type. A strong correlation was observed between *CLEC2D* expression and NK cell infiltration in HPV-negative HNSCC, and the combination of both parameters distinguished prognostic subgroups. However, it should be taken into account that *CLEC2D* mRNA levels may not strictly match with LLT1 protein expression/function and does not allow to distinguish between the tumor and the surrounding immune microenvironment. Furthermore, GSEA analysis showed that *CLEC2D*^high^ HNSCC were functionally related to more tumorigenic phenotypes than *CLEC2D*^low^ tumors, thus reinforcing a potential role of LLT1 as a novel target for immunotherapy in HNSCC patients.

On the other hand, our data unveil a possible link between LLT1/CLEC2D expression and HPV infection. In line with this, *CLEC2D* expression was found to be upregulated by HPV E6 oncoprotein [23]. Similarly, it has also been demonstrated that PD-L1 expression is induced by HPV16 E7 oncoprotein, thereby leading to antitumor response inhibition in cervical cancer [24]. HPV E7 has also been found to modulate lymphocyte proliferation and cytotoxicity [24], which may contribute to circumvent immune response. Moreover, lymphocyte-mediated cytotoxicity by CD4^+^ T cells has been involved in the control and clearance of viral infections [25]. There is also evidence for the involvement of PD-1/PD-L1 in immune resistance in HPV-positive HNSCC, which are tumors highly infiltrated by TIL and with more favorable clinical outcomes than HPV-negative HNSCC [26]. It has been pointed that PD-1/PD-L1 may play a role in HPV-related HNSCC to create an immune-privileged site, supporting both initial viral infection persistence and subsequent immune resistance during malignant progression [26]. Analogously, a similar function could plausibly apply to LLT1/CLEC2D in these tumors, which merits further future investigation.

In conclusion, this study uncovers the frequent LLT1/*CLEC2D* expression in HPV-negative OPSCC, unprecedentedly identified as an independent poor prognostic factor specifically for this cancer subtype. Accordingly, tumoral LLT1 expression could act as an immune escape mechanism in these tumors, and hence emerge as a novel immunotherapeutic target for HPV-negative HNSCC patients.

## 4. Materials and Methods

### 4.1. Patients and Tissue Specimens

Surgical tissue specimens from 249 patients with OPSCC who underwent resection of their tumors at the Hospital Universitario Central de Asturias between 1990 and 2009 were retrospectively collected. Experimental procedures were performed in accordance with the Declaration of Helsinki. Written informed consent was obtained from all patients. Formalin-fixed paraffin-embedded (FFPE) tissue samples and data from donors were provided by the Principado de Asturias BioBank (PT17/0015/0023), integrated in the Spanish National Biobanks Network, and histological diagnosis was confirmed by an experienced pathologist. Samples were processed following standard operating procedures with the appropriate approval of the Ethical and Scientific Committees of the Hospital Universitario Central de Asturias and the Regional CEIm from Principado de Asturias (date of approval 14 May 2019; approval number: 141/19, for the project PI19/00560).

All patients had a single primary tumor, microscopically clear surgical margins, and received no treatment prior to surgery. The stage of the tumors was determined according to the TNM system of the International Union Against Cancer (7th Edition). Overall, 167 (67%) of 249 patients received postoperative radiotherapy. Patients were followed-up for a minimum of 24 months. Recurrence was defined as relapse of the tumor in the five first years after treatment at any site: local recurrence, nodal metastasis, or distant metastasis. The mean follow-up for the whole series was 33 ± 36 months (median, 20 months). The mean follow-up of the patients with recurrence was 19.6 ± 16 months (median. 15 months), and the mean follow-up of the patients without tumor recurrence was 54.7 ± 47 months (median, 48 months). Information on HPV status was available for all the patients. HPV status was analyzed using p16-immunohistochemistry, high-risk HPV DNA detection by in situ hybridization and genotyping by GP5+/6+-PCR, as previously reported [27]. Eight cases that were HPV-positive were excluded from analysis.

### 4.2. Tissue Microarray (TMA) Construction

Three morphologically representative areas were selected from each individual tumor paraffin block. Subsequently, three 1 mm cylinders were taken to construct tissue microarray (TMA) blocks, as described previously [27], containing a total of 249 oropharyngeal squamous cell carcinomas (OPSCC): 141 tonsillar and 108 base of tongue carcinomas. In addition, each TMA included three cores of normal epithelium as an internal negative control. This normal epithelium was obtained from adult male, non-smokers, and non-drinkers, patients that were operated from tonsillectomy due to chronic tonsillitis.

### 4.3. Immunohistochemical Study

The TMAs were cut into 3-μm sections and dried on Flex IHC microscope slides (Dako, Glostrup, Denmark). The sections were deparaffinized with standard xylene and hydrated through graded alcohols into water. Antigen retrieval was performed by heating the sections using Envision Flex Target Retrieval solution, low pH (Dako, Glostrup, Denmark). Staining was done at room temperature on an automatic staining workstation (Dako Autostainer Plus) with mouse monoclonal antibody anti-LLT1/CLEC2D (Clone 4C7 # H00029121-M01; Novus Biologicals, Littleton, CO, USA) at 1:200 dilution. Immunodetection was carried out with the Dako EnVision Flex + Visualization System (Dako Autostainer, Denmark), using diaminobenzidine as a chromogen. Counterstaining with hematoxylin was the final step.

LLT1 immunostaining was preferentially detected in the cytoplasm of tumor cells, although some cases showed protein enrichment at the cell membrane. Expression levels were quantitated, independently, by two investigators. Since LLT1 expression was homogeneous within the tumor samples, a semiquantitative scoring system based on staining intensity was applied: LLT1 expression was classified as negative (absence of expression), weak to moderate staining, or strong staining. For statistical purposes, and based on preliminary analyses, the score was dichotomized as negative expression versus positive (weak-moderate/strong staining) expression.

LLT1 immunostaining in TIL was scored semiquantitatively, by measuring the densities of positive immune cells, as previously described [28]: no, or sporadic positive cells (score 0); mild to moderate numbers of positive cells (score 1); abundant (score 2) and highly abundant positive cells (score 3). LLT1 immunostaining in TIL was jointly evaluated in the intratumoral compartment and the stromal compartment, and scores were dichotomized as low expression (scores 0 and 1) versus high expression (scores 2 and 3).

### 4.4. In Silico Analysis of CLEC2D mRNA Expression Using The Cancer Genome Atlas (TCGA) HNSCC Database

mRNA expression analysis was performed using the transcriptome data from a TCGA cohort of 523 HNSCC patients [14]. A subset of 488 TCGA HNSCC patients was used to assess the impact of CLEC2D expression on patient survival using the platform cBioportal [29]. *CLEC2D* mRNA levels were compared between primary tumors (n = 520) and normal tissue samples (n = 44) using the UALCAN web tools (http://ualcan.path.uab.edu/) [30]. Correlations with patient survival and immune infiltration assessment were performed through TIMER 2.0 web server resource [14]. MCP-counter model was used to study the relationship between CLEC2D expression and NK cell immune infiltration [16]. 

### 4.5. Statistical Analysis

Chi-squared and Fisher’s exact tests were used for comparison between categorical variables. For time-to-event analysis, Kaplan-Meier curves were plotted. Cox proportional hazards models were utilized for univariate and multivariate analyses. The hazard ratios (HR) with 95% confidence interval (CI) and *p* values were reported. All tests were two-sided. *p* values of ≤0.05 were considered statistically significant.

## Figures and Tables

**Figure 1 biomedicines-08-00535-f001:**
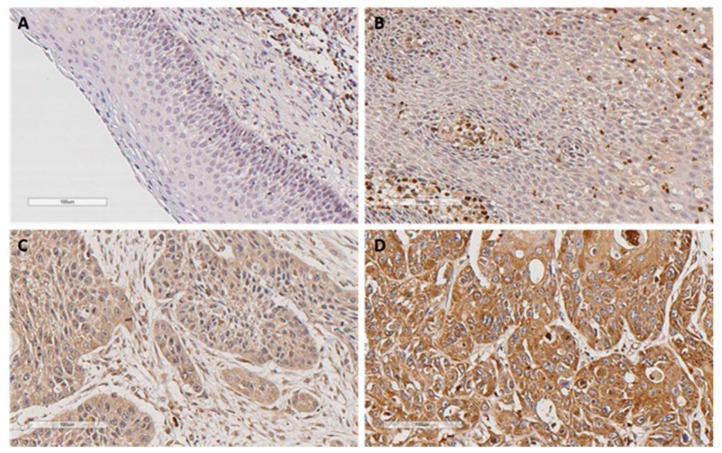
Immunohistochemical analysis of LLT1 expression in oropharyngeal squamous cell carcinomas (OPSCC). (**A**) Absence of LLT1 expression in normal epithelium. (**B**) Negative LLT1 expression in tumor cells (tumor infiltrating lymphocytes with strong LLT1 expression are also observed). (**C**) Weak to moderate LLT1 expression. (**D**) Strong LLT1 expression. Scale bar 100 µm.

**Figure 2 biomedicines-08-00535-f002:**
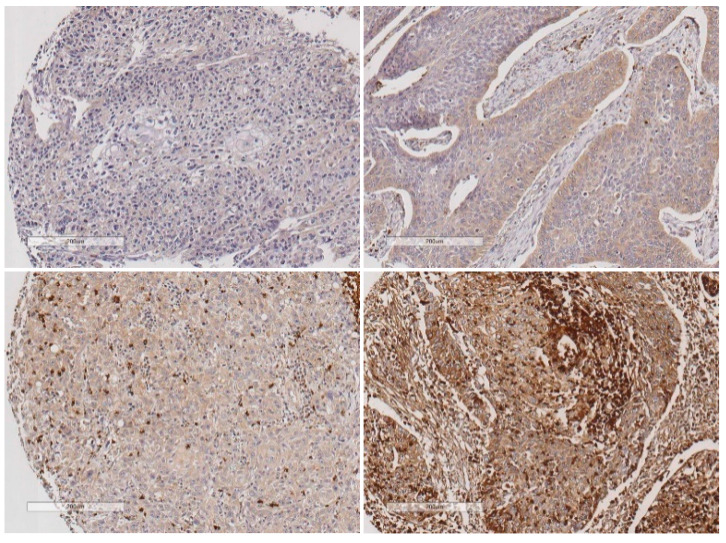
Representative examples of different density of LLT1-positive tumor-infiltrating lymphocytes, as follows: absence of infiltration (upper left panel), mild infiltration (upper right panel), abundant infiltration (lower left panel), and highly abundant infiltration (lower right panel). Scale bar 200 µm.

**Figure 3 biomedicines-08-00535-f003:**
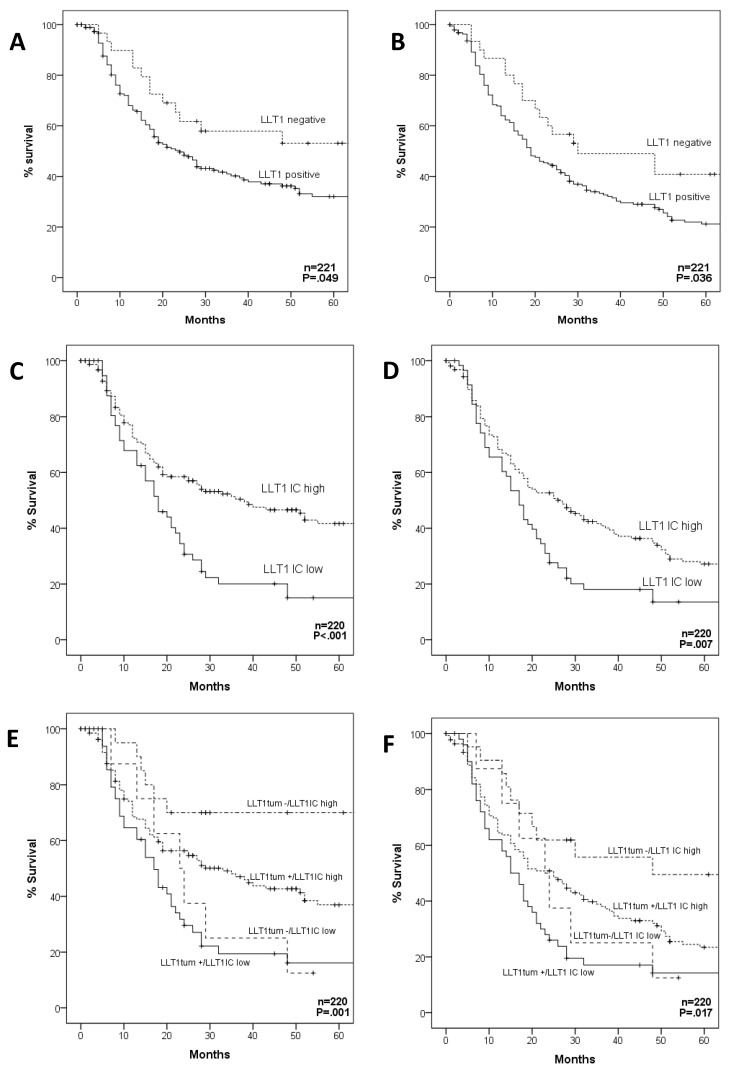
Kaplan-Meier disease-specific survival (left panels) and overall survival (right panels) curves categorized according to tumor LLT1 expression (**A**,**B**), density of LLT1-positive tumor-infiltrating lymphocytes (TIL) (LLT1 IC; (**C**,**D**)), and the combination of both parameters (**E**,**F**). *p* values estimated by log-rank test.

**Figure 4 biomedicines-08-00535-f004:**
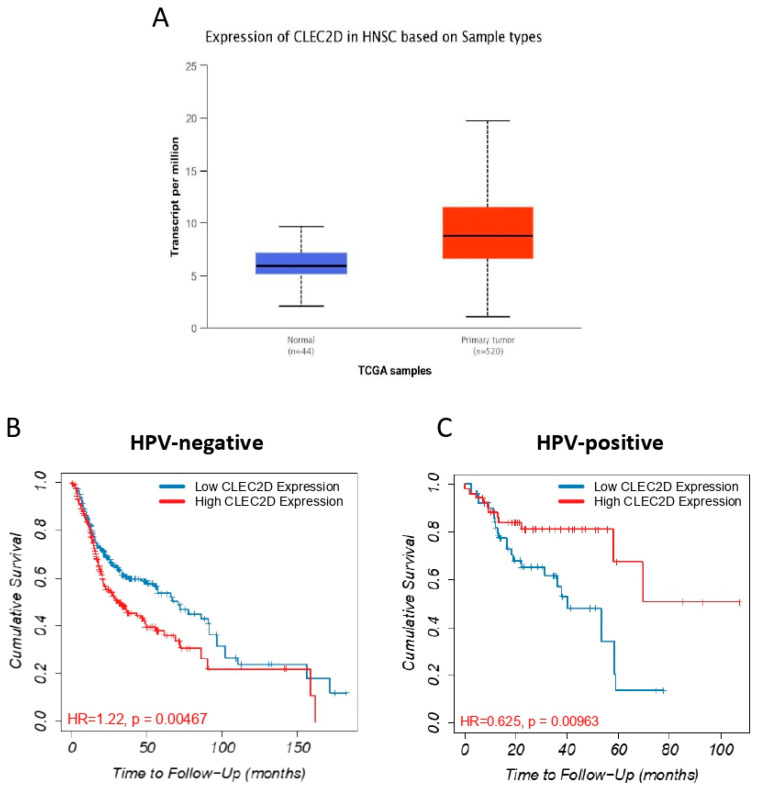
Expression and prognostic relevance of *CLEC2D* gene expression in TCGA head and neck squamous cell carcinoma (HNSCC) samples. (**A**) Box plot illustrating the comparison of *CLEC2D* mRNA levels in normal versus primary HNSCC. Kaplan-Meier curves showing overall survival categorized by *CLEC2D* mRNA expression for the subgroups of HPV-negative (**B**) and HPV-positive HNSCC (**C**).

**Figure 5 biomedicines-08-00535-f005:**
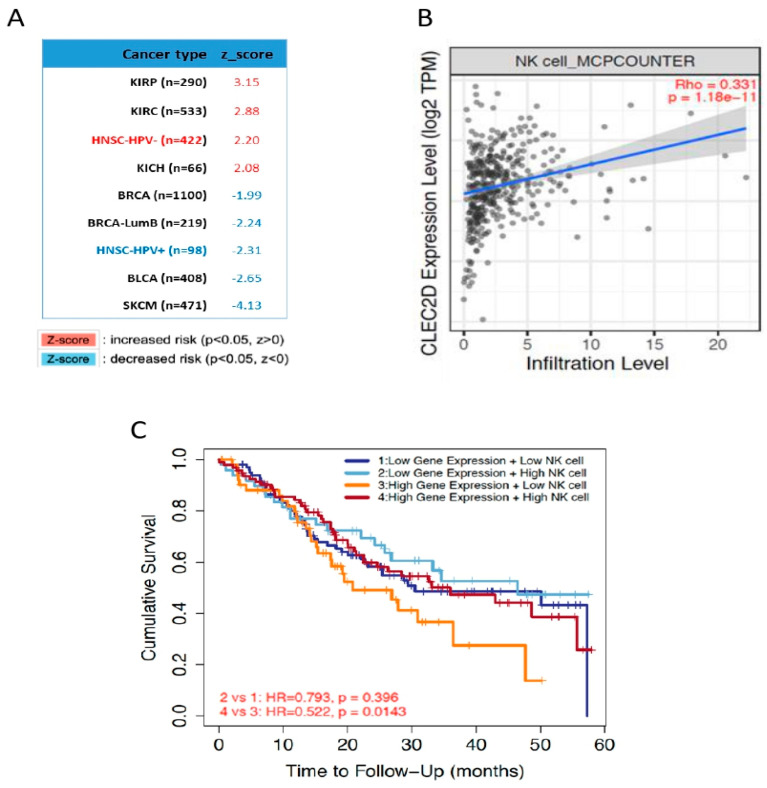
Relationship of *CLEC2D* gene expression with patient prognosis and immune infiltration. (**A**) Systematic analysis of *CLEC2D* mRNA expression in patient outcome across different cancer types. Abbreviations: KIRP, kidney renal papillary cell carcinoma; KIRC, kidney renal clear cell carcinoma; HNSC HPV-, HPV-negative head and neck squamous cell carcinoma, KICH, kidney chromophobe; BRCA, breast invasive carcinoma; BRCA-LumB, breast cancer luminal B subtype; HNSC HPV+, HPV-positive head and neck squamous cell carcinoma, BLCA, bladder cancer; SKCM, skin cutaneous melanoma. (**B**) Scatter plot for the correlation between *CLEC2D* mRNA levels and natural killer (NK) cell infiltration using MCP-counter model. (**C**) Kaplan-Meier curves of 5-year overall survival for the combination of *CLEC2D* gene expression (top 40% vs. bottom 40%) with NK cell infiltration (top 50% vs. bottom 50%).

**Figure 6 biomedicines-08-00535-f006:**
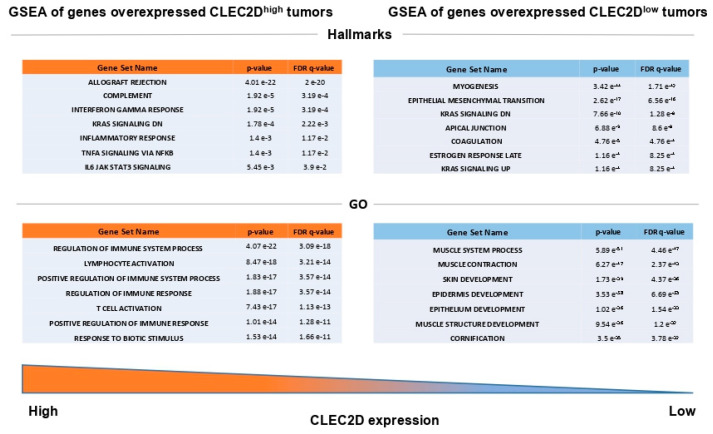
Gene set enrichment analysis (GSEA) of genes found overexpressed in *CLEC2D*^high^ and *CLEC2D*^low^ tumors. “Hallmark” and “Gene ontology biological process” (GO) data sets are shown.

**Table 1 biomedicines-08-00535-t001:** Associations between LLT1 expression and the clinicopathological characteristics of the OPSCC patients studied.

Characteristic	No. Cases (%)	LLT1-Positive Expression (%)	*p*
Age, mean (range)	57.94 (30.4–85.4 years)		
Gender			0.59
Male	214 (97%)	183 (85%)
Female	7 (3%)	7 (100%)
Tobacco			0.53
No	4 (2%)	4 (100%)
<50 Pack-year	114 (51%)	96 (84%)
>50 Pack-year	99 (45%)	87 (88%)
Unknown	4 (2%)	3 (75%)
Alcohol			1
No	4 (2%)	4 (100%)
Yes	213 (96%)	183 (86%)
Unknown	4 (2%)	3 (75%)
pT classification			0.43
T1	12 (5%)	11 (92%)
T2	53 (24%)	43 (81%)
T3	77 (35%)	64 (83%)
T4	79 (36%)	72 (91%)
pN classification			0.99
N0	52 (23%)	45 (86%)
N1	28 (13%)	24 (86%)
N2	110 (50%)	94 (85%)
N3	31 (14%)	27 (87%)
Stage			0.3
I	2 (1%)	2 (100%)
II	18 (8%)	13 (72%)
III	38 (17%)	32 (84%)
IV	163 (74%)	143 (88%)
Degree of differentiation			0.64
Well-differentiated	99 (45%)	87 (88%)
Moderately-differentiated	83 (37%)	69 (83%)
Poorly-differentiated	39 (18%)	34 (87%)
Tumor recurrence			0.1
No	85 (38.5%)	69 (81%)
Yes	136 (61.5%)	121 (89%)
Follow-up			0.1
Alive without disease	48 (21%)	36 (75%)
Dead by the disease	123 (56%)	110 (89%)
Dead by other causes	50 (23%)	44 (88%)
Total	221	190 (86%)

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
