# Peer review of "Lectin-Like Transcript 1 (LLT1) Checkpoint: A Novel Independent Prognostic Factor in HPV-Negative Oropharyngeal Squamous Cell Carcinoma"

_biomedicines, 2020, doi:10.3390/biomedicines8120535_

Round 1
Reviewer 1 Report
Head and neck squamous cell carcinoma is the most frequent tumor that arise at various locations (larynx, hypo pharynx, oropharynx and oral cavity). Many different factors play a role in the development of HNSCC.
Lectin-like transcript 1 is a ligand of NK cell inhibitory receptor NKRP1a. Several studies have reported higher expression o LLT1 in various tumors such as breast cancer and prostate cancer. Expression of LLT1 in HNSCC has not yet been investigated. This study is the first.
The Authors analyzed LLT1 expression in HPV negative, HNSCC patients. HPV plays an important role in the pathogenesis of SCC. Maybe it would be worth to justify in Introduction why the Authors included only HPV negative patients in their research.
I have no more comments. The research is reliable and may be published.
This study sheds new light on the possible use LLT1 as a biomarker that may be also be an effective immunotherapeutic target for treatment HPV negative patients with HNSCC.
Author Response
Point 1: The Authors analyzed LLT1 expression in HPV negative, HNSCC patients. HPV plays an important role in the pathogenesis of SCC. Maybe it would be worth to justify in Introduction why the Authors included only HPV negative patients in their research.
Response 1: As we mention in the Introduction, HNSCC is indeed a complex and heterogeneous disease and the importance of focusing analysis on homogeneous cohorts of patients. On this basis, we selected for study a large homogeneous cohort of HNSCC patients homogeneously treated (surgery) from a single tumor subsite (oropharynx), and similar etiology (i.e. HPV-negative). As we previously reported, the incidence of HPV infection in northern Spain is very low (6%) (Rodrigo et al. Int J Cancer. 2014 Jan 15;134(2):487-92. doi: 10.1002/ijc.28355). Therefore, there is a low number of HPV-positive HNSCC patients available to reach an adequate sample size.
I have no more comments. The research is reliable and may be published.
This study sheds new light on the possible use LLT1 as a biomarker that may be also be an effective immunotherapeutic target for treatment HPV negative patients with HNSCC.
Response: We thank the reviewer for the positive comments highlighting the quality and added value of our work and also his/her recommendation for publication.
Reviewer 2 Report
The manuscript "LECTIN-LIKE TRANSCRIPT 1 (LLT1) CHECKPOINT: A NOVEL INDEPENDENT PROGNOSTIC FACTOR IN HPV-NEGATIVE OROPHARYNGEAL SQUAMOUS CELL CARCINOMA" by Mario Sanchez-Canteli et al., represents a descriptive study of LLC1 expression in OPSCC and its prognostic impact.
In general the manuscript is well written, of interrest for the scientific community working in this specific field and represent another puzzle piece, which will help to advance and improve our understanding of the underlying patho-physiological, molecular as well as immunological mechanisms.
However a few points need to be addressed by the authors:
1) Line 61/62 NKRP1A is referred to as an inhibitory receptor, in line 64 as a potentially activating receptor, this needs more explanation. It should be also clearly stated that there are apparently controversial and most likely context dependent findings. In particular regarding its respective role in NK, B- and T-cells.
2) Line 64, "NKRP1A and CD161" these are two alternative names for the same protein and not two different proteins, please correct.
3) Line 124-126 "By contrast, high CLEC2D mRNA levels were significantly associated with a better prognosis in HPV-positive OPSCC patients", this needs an extensive discussion in the discussion section as well as inclusion of the role of CLEC2D during viral infections (please include references).
4) Please make sure that labels in the figures are large enough to be legible, keep them consistent throughout the manuscript with same letter size and font (see e.g. Fig. 3.c).
5) It is interresting that the authors find a correlation between CLEC2D expression levels and NK cell infiltration. As in the introduction this needs an extensive discussion in the discussion section, looking both into the HPV positive and negative population.
6) It would significantly strengthen the paper if the authors could provide an IHC-based CD161 analysis of the tumor collection, which they have evaluated by LLT1 staining as a receptor is nothing without the corresponding ligand and vice versa.
7) Form line 167/168 and 183/184, it appears that expression of LLT1 on immune cells or tumor cells respectively may have a differential prognostic value. The authors should elaborate more on this finding in the light of published work and furthermore analyse their own data for LLC1 staining on both cell types.
Author Response
In general the manuscript is well written, of interest for the scientific community working in this specific field and represent another puzzle piece, which will help to advance and improve our understanding of the underlying patho-physiological, molecular as well as immunological mechanisms.
Response: We thank the reviewer for the positive comments and also for his/her valuable suggestions to improve our manuscript.
However a few points need to be addressed by the authors:
Point 1: Line 61/62 NKRP1A is referred to as an inhibitory receptor, in line 64 as a potentially activating receptor, this needs more explanation. It should be also clearly stated that there are apparently controversial and most likely context dependent findings. In particular regarding its respective role in NK, B- and T-cells.
Response 1: Even though a dual role has been proposed for LLT1/CD161 complex (Aldemir et al. J Immunol 2005; 175:7791-7795, doi: 10.4049/jimmunol.175.12.7791), to our knowledge there is functional evidence only supporting an inhibitory role. To avoid confusion this sentence has been rephrased in this revised version.
Point 2: Line 64, "NKRP1A and CD161" these are two alternative names for the same protein and not two different proteins, please correct.
Response 2: We fully agree. This has now been corrected.
Point 3: Line 124-126 "By contrast, high CLEC2D mRNA levels were significantly associated with a better prognosis in HPV-positive OPSCC patients", this needs an extensive discussion in the discussion section as well as inclusion of the role of CLEC2D during viral infections (please include references).
Response 3: The possible link between CLEC2D expression and HPV infection has now been discussed more extensively, as suggested.
Point 4: Please make sure that labels in the figures are large enough to be legible, keep them consistent throughout the manuscript with same letter size and font (see e.g. Fig. 3.c).
Response 4: Figures are now consistently labeled.
Point 5: It is interresting that the authors find a correlation between CLEC2D expression levels and NK cell infiltration. As in the introduction this needs an extensive discussion in the discussion section, looking both into the HPV positive and negative population.
Response 5: The correlation between CLEC2D expression and NK cell infiltration has now been discussed. We only showed expression data and survival analysis for the HPV-negative subgroup, matching the patient cohort used for LLT1 analysis.
Point 6: It would significantly strengthen the paper if the authors could provide an IHC-based CD161 analysis of the tumor collection, which they have evaluated by LLT1 staining as a receptor is nothing without the corresponding ligand and vice versa.
Response 6: Noteworthy, this study is the first to investigate the significance of LLT1/CLEC2D expression in HNSCC, unprecedentedly revealed as an independent prognostic factor. CD161 receptor could also be clinically and biologically relevant for this complex, and as such it merits further future investigation. It should also be taken into consideration that completion of CD161 analysis extends beyond the deadline for this revision.
Point 7: Form line 167/168 and 183/184, it appears that expression of LLT1 on immune cells or tumor cells respectively may have a differential prognostic value. The authors should elaborate more on this finding in the light of published work and furthermore analyse their own data for LLC1 staining on both cell types.
Response 7: Following the reviewer’s suggestion, LLT1 expression in tumor-infiltrating lymphocytes was also quantified and extensive new data and figures have been included in this revised version of the manuscript, and also further discussed.
Reviewer 3 Report
The research paper by Sanchez-Canteli M. et al., represents a clearly written and concise overview of data that validate the negative prognostic value of LLT1 in HPV- OPSCC. Next to over 200 human samples, also data from TCGA were used for confirmation in OPSCC and other cancers with different. A paper with important information for the antitumor immunotherapy field.
The only minor comment I would like to make is that the abbreviation list found in Figure 4A should be written in full in the figure legend. Plus the title of the legend of Figure 4 states:' Relationship of CLEC2D gene expression with patient prognosis and immune infiltration'. --> I like to suggest to replace 'immune' by 'NK cell'. NK cell infiltration can not be mistaken for immune infiltration (by both innate and adaptive myeloid and lymphoid players).
Author Response
The research paper by Sanchez-Canteli M. et al., represents a clearly written and concise overview of data that validate the negative prognostic value of LLT1 in HPV- OPSCC. Next to over 200 human samples, also data from TCGA were used for confirmation in OPSCC and other cancers with different. A paper with important information for the antitumor immunotherapy field.
Response: We thank the reviewer for his/her positive comments highlighting that our work is important for the antitumor immunotherapy field.
Point 1: The only minor comment I would like to make is that the abbreviation list found in Figure 4A should be written in full in the figure legend. Plus the title of the legend of Figure 4 states:' Relationship of CLEC2D gene expression with patient prognosis and immune infiltration'. --> I like to suggest to replace 'immune' by 'NK cell'. NK cell infiltration can not be mistaken for immune infiltration (by both innate and adaptive myeloid and lymphoid players).
Response 1: The title and abbreviations in Legend for Figure 4 have been modified, according to the reviewer’s recommendation.